# Differential protein expression of GABA A receptor alpha 1 subunit and calbindin in rat spermatozoa associated with proteomic analysis in testis following methamphetamine administration

Paweena Kaewman[1,2] , Sutisa Nudmamud-Thanoi[2,3] *, Jitnapar Thongleart[2], Sawanya Charoenlappanit[4], Sittiruk Roytrakul[4], Samur Thanoi[5] *

1 School of Medicine, Walailak University, Nakhon Si Thammarat, Thailand, 2 Department of Anatomy, Faculty of Medical Science, Naresuan University, Phitsanulok, Thailand, 3 Centre of Excellence in Medical Biotechnology, Naresuan University, Phitsanulok, Thailand, 4 Functional Ingredients and Food Innovation Research Group, National Center for Genetic Engineering and Biotechnology, National Science and Technology Development Agency, Pathum Thani, Thailand, 5 School of Medical Sciences, University of Phayao, Phayao, Thailand

These authors contributed equally to this work.

* sutisat@nu.ac.th (SNT); samur.t@up.ac.th (ST)

**Data Availability Statement:** All relevant data are within the manuscript.

## Abstract

Methamphetamine (METH) can induce spermatogenesis impairment, testicular apoptosis, and abnormal sperm quality. It also promotes changes in the expression of receptors for sex hormones and neurotransmitters, including GABA receptors in the testis. Proteomic assessment focusing on proteins involved in the calcium signalling pathway in the testis can facilitate diagnostic factors contributing to testicular and sperm functions, especially those related to spermatogenesis and fertilisation. In this study, we proposed to determine the localisation and differential expression of GABA A receptor alpha 1 subunit (GABA A-α1) in the spermatozoa of METH-administered rats. The differential proteomic profile of the testis was also observed by focusing on proteins in the KEGG pathways belonging to the calcium signalling pathway. There were 212 differentially expressed proteins in the rat testis, based on the cut-off value of 1.2-fold change. Most of those proteins, 13 proteins, were classified in the calcium signalling pathway, including 4 down-regulated and 9 up-regulated proteins. An immunolocalisation study of the GABA A-α1 receptor and calbindin revealed their localisation in the equatorial segment of the head in the rat spermatozoa. The expression of calbindin is also found in the middle piece of sperm. An increase in GABA A-α1 receptor in rat spermatozoa was correlated with an increase in abnormal sperm motility and morphology after methamphetamine exposure. Moreover, calbindin expression in sperm decreased in METH-administered rats. All our findings demonstrate that METH influences intracellular calcium homeostasis by acting through the calcium signalling pathway-associated proteins. Moreover, it might disrupt ion homeostasis in sperm through the GABA A-α1 receptor and calbindin, triggering a change in intracellular calcium and chloride ions. These changes may

**Funding:** This study was funded by the Naresuan University Research Fund (R2561B016 and R2565B043). The funders had no role in study design, data collection and analysis, decision to publish, or preparation of the manuscript.

**Competing interests:** The authors have declared that no competing interests exist.

cause abnormalities in spermatogenesis, testicular apoptosis, and sperm quality impairment.

## Introduction

Methamphetamine (METH), an illicit drug, contributes to approximately 35 million METH abusers worldwide [1]. METH is one of many addictive drugs with known adverse effects on the reproductive system. Several studies found that apoptotic activity was significantly increased in the testes of rodents given METH [2–4]. METH affects not only the spermatogenic cells in the testis but also the epididymal sperm. In 2016, Nudmamud-Thanoi et al. revealed a significant decrease in the percentage of normal sperm motility and morphology in METH-administered rats, notably in the escalating dose-binge group, which mimicked the most extensively used methods for METH usage in humans. A significant decrease in progesterone and oestrogen receptor immunostaining in Sertoli cells and spermatogenic cells was reported in the seminiferous tubules of those rats [5]. It is interesting to note that after exposure to METH, alterations in receptors for gamma-aminobutyric acid (GABA), dopamine, and catecholamines in the testis were also found [6, 7].

Many studies have revealed that GABA participates in testicular functions such as spermatogenesis, testosterone production, and proliferation of spermatogonial stem cells and Leydig cells by acting through GABA receptors and transporters [8–11]. Moreover, GABA receptors play a role in sperm function (i.e., sperm motility, capacitation, hyperactivation, and acrosome reaction) through control of calcium and potassium channel opening, calcium ion influx, and cyclic 3', 5' adenosine monophosphate (cAMP) accumulation [12–19]. The GABA A receptor is a ligand-gated chloride channel receptor involved in cell membrane hyperpolarisation [17]. By acting through GABA A receptors in the sperm head, GABA can promote changes in cAMP, intracellular calcium and chloride ions, and protein tyrosine phosphorylation [18, 19]. This evidence suggests that GABA A receptors might also be involved in sperm quality responding to METH exposure, but the physiological function remains unknown.

Our previous study in the testicular proteome of METH-administered rats reported differentially expressed proteins involving signal transduction [20]. The testicular protein profiles in the condition of abnormal testicular function were limited. Therefore, we are interested in expanding on this research in order to obtain a more complete view of the specific proteins involved in the impairment of testicular function and sperm quality. The modulation of the proteomic profiles in rat testis induced by METH was analysed by integrating the KEGG database. To better understand the impact of the GABA A receptors on sperm quality, the expression of GABA A receptor alpha 1 subunit (GABA A-α1) in mature spermatozoa and its correlation with sperm quality parameters were identified in this study.

## Materials and methods

### Animals and METH administration

All animal procedures and METH administration methods were carried out in compliance with the Guide for the Care and Use of Laboratory Animals (National Research Council of Thailand), as previously described [6]. The protocols in this study were approved by the ethical committee of Naresuan University, Thailand (Naresuan University Animal Care and Use Committee (NUACUC)) (Approval numbers: 60 02 004 and 62 02 012). For daily animal care procedures, male Sprague-Dawley rats were given unrestricted access to food and water and

housed in cages with a 12:12 hour light/dark cycle at 24±1˚C. For proteomic analysis, there were 2 groups of rats (n = 8–9 per group), including control and escalating dose-binge (ED-binge METH) groups. The rats in the control group were administered 0.9% normal saline by intraperitoneal (IP) injection for 15 days, while the rats in the ED-binge METH group were IP-injected for 14 days with gradually increasing doses of METH (0.1–4 mg/kg), followed by a binge dose of METH (6 mg/kg METH, four times a day at every 2 hours) on day fifteen. We also examined the expression of the GABA A-α1 receptor in spermatozoa of the rat in those groups and in the acute binge (AB METH) group (n = 8). The rats in AB METH group were IP-injected with 0.9% normal saline for 14 days and with a binge dose of METH on day fifteen. After the last injection of the treatment, the rats were sacrificed by cervical dislocation, which was performed by a trained and experienced technician. Conscious behaviours of the rats, including breath holding, movement, and pedal withdrawal reflex, were carefully observed for verification of death. Exsanguination through cardiac puncture was performed as a secondary physical method of euthanasia.

## Sample collection

The testis was promptly removed after the sacrifice at the end of the treatment and then frozen at -80˚C. The cauda epididymis was dissected and minced in warm phosphate buffer saline (PBS, pH 7.4) for sperm extraction. To evaluate sperm quality and GABA A-α1 receptor expression, epididymal sperm were extracted. Sperm quality, including sperm concentration, motility, and morphology, was analysed in accordance with the method mentioned previously [5].

## Liquid chromatography–tandem mass spectrometry (LC–MS/MS)

The proteomic analysis of testicular proteins has been previously described by Thongleart [20]. Testicular proteins were extracted from 30 mg of frozen testis tissue by homogenising in cold homogenising buffer (25 mM NaCl in 5 mM Tris-HCl, pH 8.0). The pellet was removed after centrifuging the homogenate at 14,000 rpm for 10 minutes. Then, it was homogenised once more in lysis buffer (50 mM Tris-HCl containing 0.15 M NaCl, 0.1% SDS, 0.25% Sodium deoxycholate, and 1% Protease inhibitor cocktail). The Pierce BCA Protein Assay kit (Thermo Fisher Scientific) was used to determine protein quantification. The proteomic analysis in each group was investigated using a pool of extracted proteins (10 μg protein) from three testis samples. For protein digestion, the samples in each pool were reduced with 10 mM dithiothreitol and alkylated with 30 mM iodoacetamide, which were incubated at 56˚C for an hour and at room temperature for an hour, respectively. Subsequently, in-solution digestion of protein was completed by incubating at 37˚C overnight with 50 ng of trypsin in 10 mM ammonium bicarbonate. All solutions for protein digestion were diluted with 10 mM ammonium bicarbonate. The eluted peptides were dried before being resuspended in 0.1% formic acid. The solution was then centrifuged at 10,000 rpm for 10 minutes and peptides in the supernatant were evaluated with the Impact II UHR-TOF MS system (Bruker Daltonics Ltd.) coupled with the nanoLC system (Thermo Fisher Scientific). The protein quantification in individual samples was conducted using Maxquant 1.6.6.0 and the Andromeda search engine to correlate MS/MS spectra to the Uniprot *Rattus norvegicus database* [21]. Label-free quantitation with Max-Quant's standard settings was performed: maximum of two miss cleavages, mass tolerance of 0.6 dalton for the main search, trypsin as digesting enzyme, carbamidomethylation of cysteine as fixed modification, and the oxidation of methionine and acetylation of the protein N-terminus as variable modifications. Only peptides with a minimum of 7 amino acids, as well as at least one unique peptide, were required for protein identification. Only proteins with at least

two peptides, and at least one unique peptide, were considered as being identified and used for further data analysis. Protein FDR was set at 1% and estimated by using the reversed search sequences. The maximal number of modifications per peptide was set to 5. Max intensities were log2 transformed, missing values were also imputed in Perseus 1.6.6.0 using constant value (zero) [22].

## Bioinformatics analysis

We used a Venn diagram to examine the effects of METH on the contents of the identified proteins and differentially expressed proteins [23]. The analysis of KEGG pathways in each category was then conducted using the Database for Annotation, Visualization, and Integrated Discovery (DAVID version 6.8, https://david-d.ncifcrf.gov/) to identify major signalling pathways involved in testicular impairment. To annotate the information of the proteins, we used the UniProt Knowledgebase (UniProt KB, https://www.uniprot.org/uniprot/) [24]. The Multi-Experiment Viewer (MeV, Version 4.9) software was used to compare protein quantification, and the results were displayed as a heat map. We utilised the STITCH 5.0 database (http://stitch.embl.de/) to predict the network of protein-chemical interactions.

## Immunocytochemistry analysis

The smearing of the epididymal sperm was done on adhesive-coated glass slides and then air-dried at room temperature. The primary antibodies used in this study were a rabbit polyclonal anti-GABA A-α1 receptor antibody (Abcam) and a rabbit polyclonal anti-calbindin (Calb1) antibody (Sigma-Aldrich). Then, the sections were incubated with a biotinylated secondary antibody, and signals were enhanced by using avidin-biotinylated horseradish peroxidase complexes (Vector). The GABA A-α1 receptor immunoreactivities were visualised using DAB (3,3'-Diaminobenzidine) (Vector). The sections were then dehydrated with serial ethanol dilutions, cleared in xylene, and mounted with histopathology mounting medium. We observed 400 sperm from two slides (200 sperm per slide) under a light microscope. The quantitative protein expression of GABA A-α1 receptor immunostaining was determined by digital image analysis tokening by a microscope camera. The intensity values of immunoreactions were measured by ImageJ software (NIH, Bethesda, MD, freely available at https://imagej.nih.gov/ij/) and represented as normalised relative optical density (ROD). The percentage of strong-positively stained sperm was calculated, which is defined as sperm with a strong positive immunoreaction.

## Statistical analysis

The Shapiro-Wilk test was used to determine the data's normal distribution. The statistical difference between two groups and three groups was analysed using unpaired Student's t-test and One–way ANOVA followed by Dunnett's post hoc test (parametric data), respectively. Statistically significant was considered at $P < 0.05$. The data are shown as mean ± SEM. Moreover, Pearson's correlation coefficient was used to investigate the correlation in all parameters.

## Results

### KEGG pathway analysis of the differentially expressed proteins in rat testis after METH exposure

A Venn diagram demonstrating the 383 differentially expressed proteins in the ED-binge METH group compared with a control group has been reported in our previous study [20]. Interestingly, 212 proteins out of a total of 383 proteins had a fold change of 1.2 ($\leq$ -1.2

**Table 1. KEGG pathways of differentially expressed proteins in rat testis (METH vs. control).**

| KEGG pathway name | Count | Percent (%) | P value | Fold enrichment |
|---|---|---|---|---|
| Calcium signalling pathway | 13 | 6.2 | 1.5E-05 | 4.8 |
| Proteoglycans in cancer | 10 | 4.8 | 5.5E-04 | 4.2 |
| Ras signalling pathway | 9 | 4.3 | 4.6E-03 | 3.4 |
| Phosphatidylinositol signalling system | 7 | 3.3 | 6.7E-04 | 6.5 |
| Inositol phosphate metabolism | 6 | 2.9 | 1.4E-03 | 7.2 |
| ErbB signalling pathway | 6 | 2.9 | 2.7E-03 | 6.2 |
| Glutamatergic synapse | 6 | 2.9 | 9.4E-03 | 4.6 |
| GnRH secretion | 5 | 2.4 | 5.9E-03 | 6.8 |
| Non-small cell lung cancer | 5 | 2.4 | 9.3E-03 | 6.0 |

and $\geq$ 1.2). Of these proteins, 111 proteins with upregulated expression and 101 proteins with downregulated expression were identified. From those proteins, nine categories of KEGG pathways with p < 0.01 were classified (Table 1). The most differentially expressed proteins in rat testis between control and ED-binge METH groups were associated with the calcium signalling pathway. Moreover, they were related to other pathways, including proteoglycans in cancer, Ras signalling pathway, phosphatidylinositol signalling system, inositol phosphate metabolism, ErbB signalling pathway, glutamatergic synapse, GnRH secretion, and non-small cell lung cancer.

The differentially expressed proteins having a fold change of 1.2 in the calcium signalling pathway were selected as the interesting proteins. Therefore, among thirteen proteins, four proteins were downregulated, including metabotropic glutamate receptor 5 (Grm5), hepatocyte growth factor (Hgf), plasma membrane calcium-transporting ATPase 1 (Atp2b1), and 1-phosphatidylinositol 4,5-bisphosphate phosphodiesterase delta-4 (Plcd4). In addition, the other nine proteins were upregulated, including 1-phosphatidylinositol 4,5-bisphosphate phosphodiesterase gamma-2 (Plcg2), 1-phosphatidylinositol 4,5-bisphosphate phosphodiesterase epsilon-1 (Plce1), leukotriene B4 receptor 2 (Ltb4r2), voltage-dependent L-type calcium channel subunit alpha-1D (Cacna1d), 1-phosphatidylinositol 4,5-bisphosphate phosphodiesterase beta-4 (Plcb4), voltage-dependent T-type calcium channel subunit alpha-1I (Cacna1i), receptor tyrosine-protein kinase erbB-2 (Erbb2), inositol 1,4,5-trisphosphate receptor type 2 (Itpr2), BDNF/NT-3 growth factors receptor (Ntrk2) (Table 2). The heat map of these proteins clarifies the difference in their quantification between the two groups (Fig 1). Most of the differentially expressed proteins of the calcium signalling pathway in rat testis were predicted to interact with METH based on the online STITCH 5.0 database (Fig 2). Although Atp2b1 and Cacna1i proteins did not show any interaction network with METH, Atp2b1 connects with this network via Calb1, which directly interacts with Ntrk2.

## GABA A-α1 receptor and calbindin expressions in rat spermatozoa

The anterior acrosomal segment (the equatorial segment) of the rat sperm's head region was shown to have the localisation of the GABA A-α1 receptor (Fig 3). The protein expression level of the GABA A-α1 receptor was assessed in the sperm head, which showed negative-to-strongly-positive immunostaining (Fig 4A–4D). The percentage of the strong-positively stained sperm for GABA A-α1 receptor immunostaining was significantly increased in METH-administered groups, AB METH (72.06 ± 5.29) and ED-binge METH (71.34 ± 4.15) groups, compared with the control group (32.19 ± 4.80) (Fig 5). Moreover, the ROD of the GABA A-α1 receptor expression on sperm was significantly increased in the AB METH

**Table 2. Differentially expressed proteins of the calcium signalling pathway in rat testis (METH vs. control).**

| No. | Protein ID | Protein name | Gene symbol | Fold change[a] (METH/Control) |
|---|---|---|---|---|
| **Down-regulation** | | | | |
| 1 | P31424 | Metabotropic glutamate receptor 5 | Grm5 | -3.82 |
| 2 | P17945 | Hepatocyte growth factor | Hgf | -2.70 |
| 3 | P11505 | Plasma membrane calcium-transporting ATPase 1 | Atp2b1 | -2.67 |
| 4 | Q62711 | 1-phosphatidylinositol 4,5-bisphosphate phosphodiesterase delta-4 | Plcb4 | -2.54 |
| **Up-regulation** | | | | |
| 5 | P24135 | 1-phosphatidylinositol 4,5-bisphosphate phosphodiesterase gamma-2 | Plcg2 | 1.20 |
| 6 | Q99P84 | 1-phosphatidylinositol 4,5-bisphosphate phosphodiesterase epsilon-1 | Plce1 | 1.47 |
| 7 | Q924U0 | Leukotriene B4 receptor 2 | Ltb4r2 | 1.80 |
| 8 | P27732 | Voltage-dependent L-type calcium channel subunit alpha-1D | Cacna1d | 1.87 |
| 9 | Q9QW07 | 1-phosphatidylinositol 4,5-bisphosphate phosphodiesterase beta-4 | Plcd4 | 2.41 |
| 10 | Q9Z0Y8 | Voltage-dependent T-type calcium channel subunit alpha-1I | Cacna1i | 2.72 |
| 11 | P06494 | Receptor tyrosine-protein kinase erbB-2 | Erbb2 | 3.23 |
| 12 | P29995 | Inositol 1,4,5-trisphosphate receptor type 2 | Itpr2 | 3.37 |
| 13 | Q63604 | BDNF/NT-3 growth factors receptor | Ntrk2 | 3.77 |

[a]Values > 0 represent up-regulation, whereas values < 0 represent down-regulation.

(2.12 ± 0.21) and ED-binge METH (2.13 ± 0.16) groups compared with the control group (1.00 ± 0.10) (Fig 6).

Calb1 is localised in the anterior acrosomal segment of the head and the middle piece of rat sperm (Fig 7). The Calb1 protein expression in the rat sperm showed negative-to-strongly-positive immunostaining (Fig 8A and 8B). The percentage of the strong-positively stained sperm for Calb1 immunostaining was significantly decreased in METH-administered groups, ED-binge METH group (2.55 ± 0.60), compared with the control group (43.95 ± 2.65) (Fig 9). Moreover, the ROD of the Calb1 expression on sperm was significantly decreased in the ED-binge METH group (0.35 ± 0.03) compared with the control group (1.00 ± 0.02) (Fig 10).

### Correlations of GABA A-α1 receptor expression in rat spermatozoa with sperm quality parameters

This study found a negative correlation between the percentage of normal sperm motility and the protein expression of GABA A-α1 receptor in sperm (ROD: r = -0.4745, p = 0.0143; the percentage of strong-positively stained sperm: r = -0.5868, p = 0.0016) (Fig 11). Moreover, the expression of GABA A-α1 receptor was negatively related to the normal form of sperm morphology (ROD: r = 0.4619, p = 0.0175; the percentage of strong-positively stained sperm: r = -0.5418, p = 0.0043) (Fig 12).

### Discussion

The proteomic studies identifying pathways related to the testicular impairment have been reported in several models for toxicity assessment, such as arsenic-induced male reproductive toxicity [25]. In this study, we have demonstrated differentially expressed proteins in the testis from METH-administered rats to evaluate which pathways mapped in the KEGG pathway are most affected after METH exposure. Our results from KEGG analysis using a 1.2-fold change cut-off indicated thirteen proteins classified in the calcium signalling pathway. Among these proteins, seven proteins have been previously classified in the signal transduction pathways by comparative analysis of biological processes, including Grm5, Plcd4, Plce1, Plcb4, Ltb4r2,

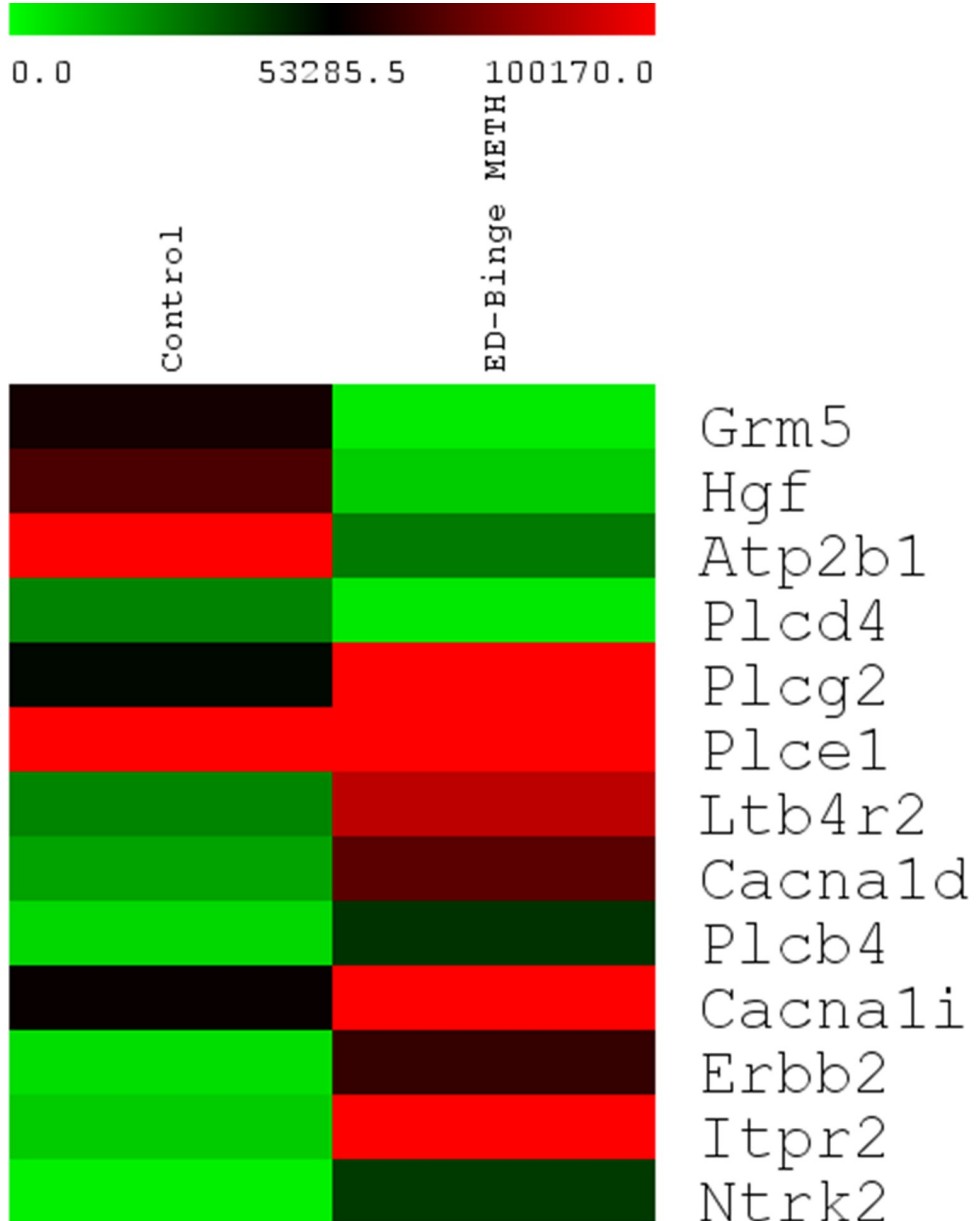

**Fig 1. A heat map representing differentially expressed proteins of the calcium signalling pathway in rat testis (METH vs. control).** Bright green, dark green, and bright red colours represent the absence, lowest, and highest protein expression, respectively.

Erbb2, and Ntrk2 [20]. The other proteins, including Hgf, Atp2b1, Plcg2, Cacna1d, Cacna1i, and Itpr2, were the first identified as differentially expressed proteins in METH-administered rat testis.

Several testicular proteins in the calcium signalling pathway such as Grm5, Hgf, Atp2b1, and Plcd 4 were down-regulated because of METH administration. On the other hand, other proteins, including Plce1, Plcb4, Plcg2, Ltb4r2, Erbb2, Ntrk2, Cacna1d, Cacna1i, and Itpr2 were up-regulated. Grm5 has been reported to be localised in several regions of the sperm, including the acrosome, midpiece, and tail. The finding of Grm5 expression in the midpiece and tail of mature sperm indicates its essentiality in sperm motility. Interestingly, it is also

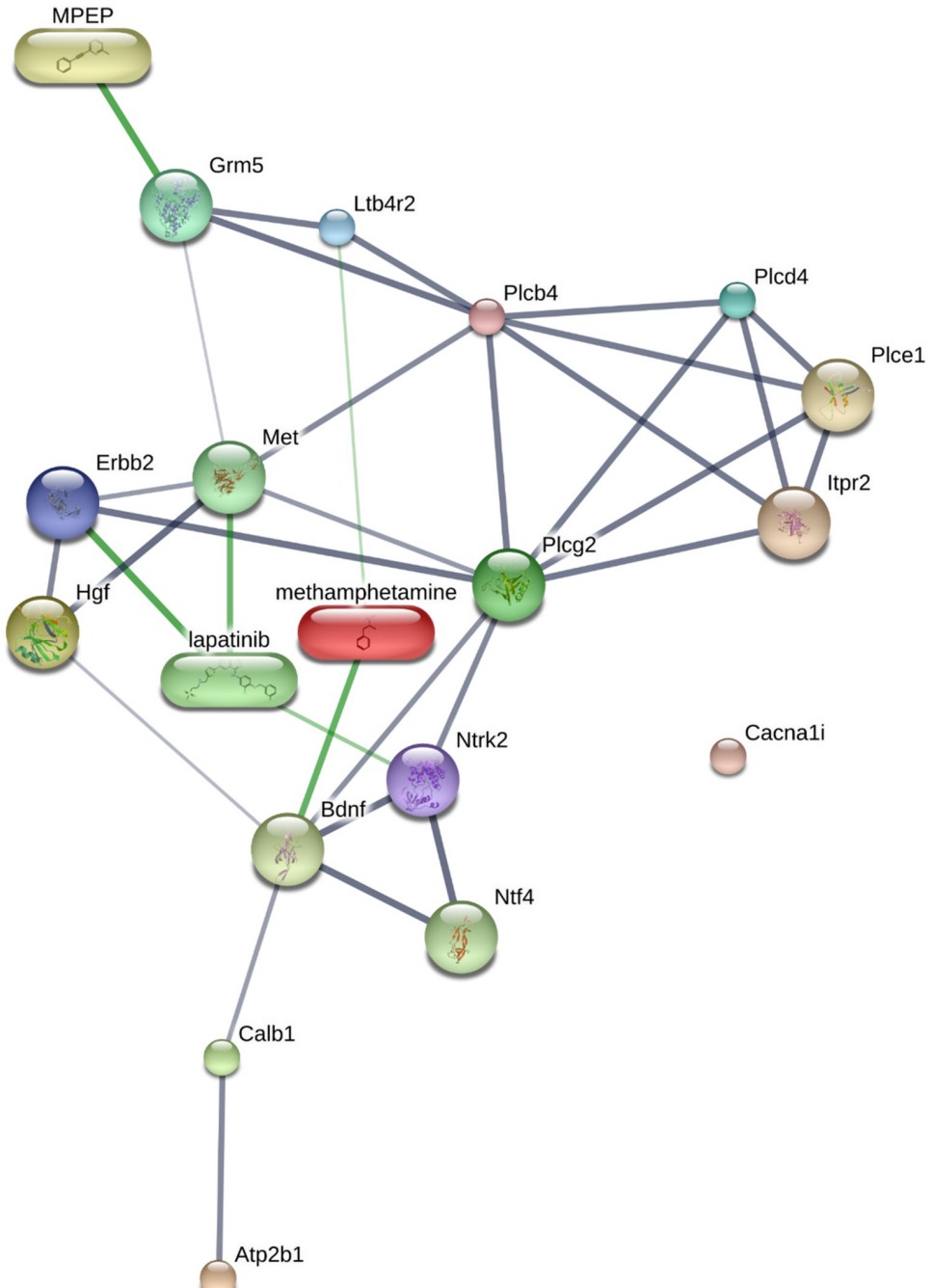

**Fig 2. Predicted interactions between METH and differentially expressed proteins of the calcium signalling pathway in rat testis.**

involved in the appearance of calcium oscillations in sperm, which is important for sperm motility and acrosome reaction [26, 27]. Hgf is synthesised by the peritubular myoid cells and Sertoli cells in the testis [28, 29]. It has been reported that its receptor is in several sex organs, including the testis, prostate gland, and ovary [30]. Its receptor, C-met, is found in both the testis and the epididymis on the head and flagellum of spermatozoa [28, 31]. Hgf plays a role in promoting a variety of biological processes occurring in the testis, such as the differentiation

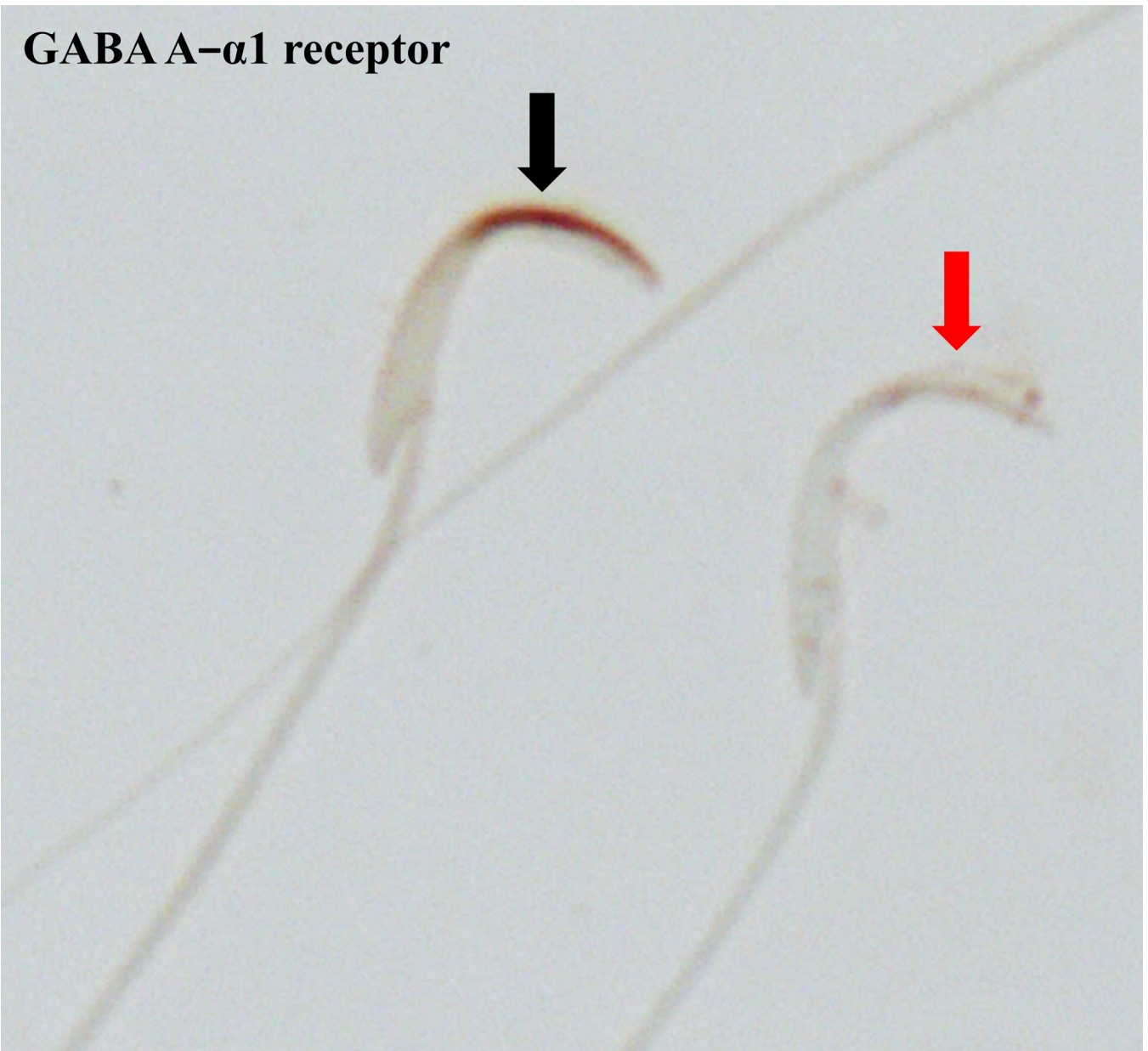

**Fig 3. Immunostaining for GABA A-α1 receptor in rat epididymal spermatozoa at 1000× magnification.** A black arrow indicates an immunopositive spermatozoon, whereas a red arrow indicates an immunonegative spermatozoon.

of testicular cells, mitosis in spermatogenesis, and apoptosis. For example, Hgf induces the proliferation of spermatogonia, the function of Leydig cells secreting testosterone hormone, and the reduction of apoptotic testicular cells in the testis [29, 31]. Moreover, it also enhances sperm maturation and sperm motility during transit through the epididymis [28, 30, 32].

As mentioned above, the present study found the alteration of four types of phospholipase C (PLC), including beta, gamma, delta, and epsilon. The delta type of PLC (Plcd) is the most well reported for its evidence associated with testis and sperm. Plcd4, a member of the Plcd, is found abundantly in spermatogenic cells of the testis, especially spermatogonia and round spermatids. Moreover, it was found in the acrosome of mature sperm and plays a role in the

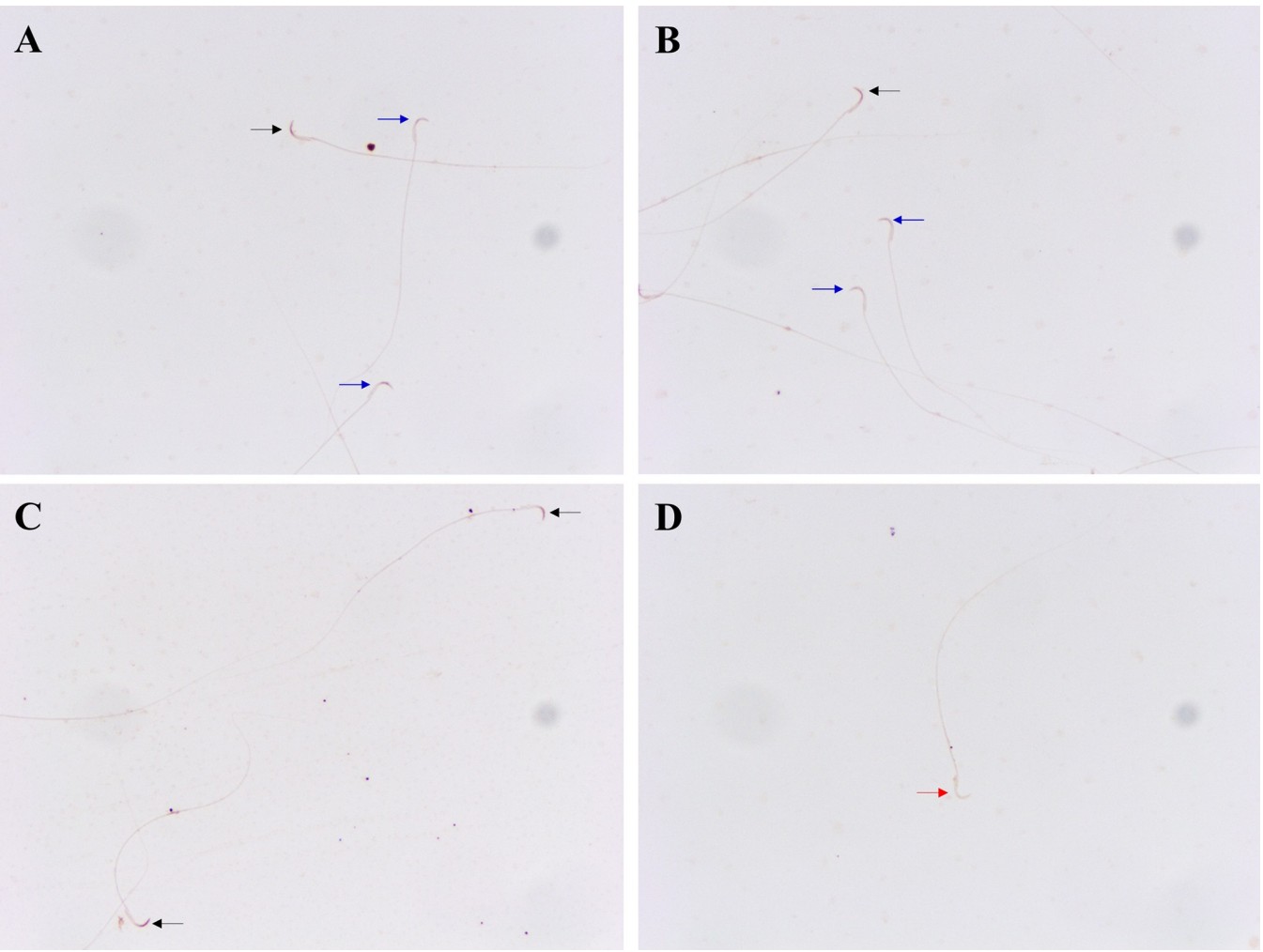

**Fig 4. Negative-to-strongly-positive immunostaining for GABA A-α1 receptor in rat epididymal spermatozoa at 200× magnification.** (A-C) Black arrows indicate the strong-positively stained sperm. (A, B) Blue arrows indicate the weak-positively stained sperm. (D) A red arrow indicates a negatively stained spermatozoon.

acrosomal formation and function. Interestingly, the impairment of fertility has been reported in the Plcd4 knockout male mice, which might be caused by the abnormality of calcium oscillations in sperm [33, 34]. We can suggest that Grm5, Hgf, and Plcd4 interact with METH exposure, leading to the impairment of testicular function. Moreover, the inositol 1,4,5-triphosphate receptor acts together with Plcd4 to regulate calcium ion channels. An increase in the inositol 1,4,5-triphosphate receptor might be present to compensate for the intracellular calcium ion concentration of sperm.

CAC1D and CAC1I are L-type and T-type voltage-dependent calcium ion channels (VDCCs), respectively. In the testis, blocking of these calcium ion channels triggers the impairment of spermatogenesis and testosterone synthesis [35]. Benoff et al. describe how L-type VDCCs are localised in testicular cells and alter calcium homeostasis, which is important in controlling apoptosis [36]. Interestingly, they are induced by follicle-stimulating hormone (FSH), resulting in calcium ion uptake following Sertoli cell proliferation and differentiation [37]. Evidence from animal models of METH exposure indicates that METH causes a decrease in FSH receptor expression in the testis [38]. Therefore, the increase of CAC1D and CAC1I

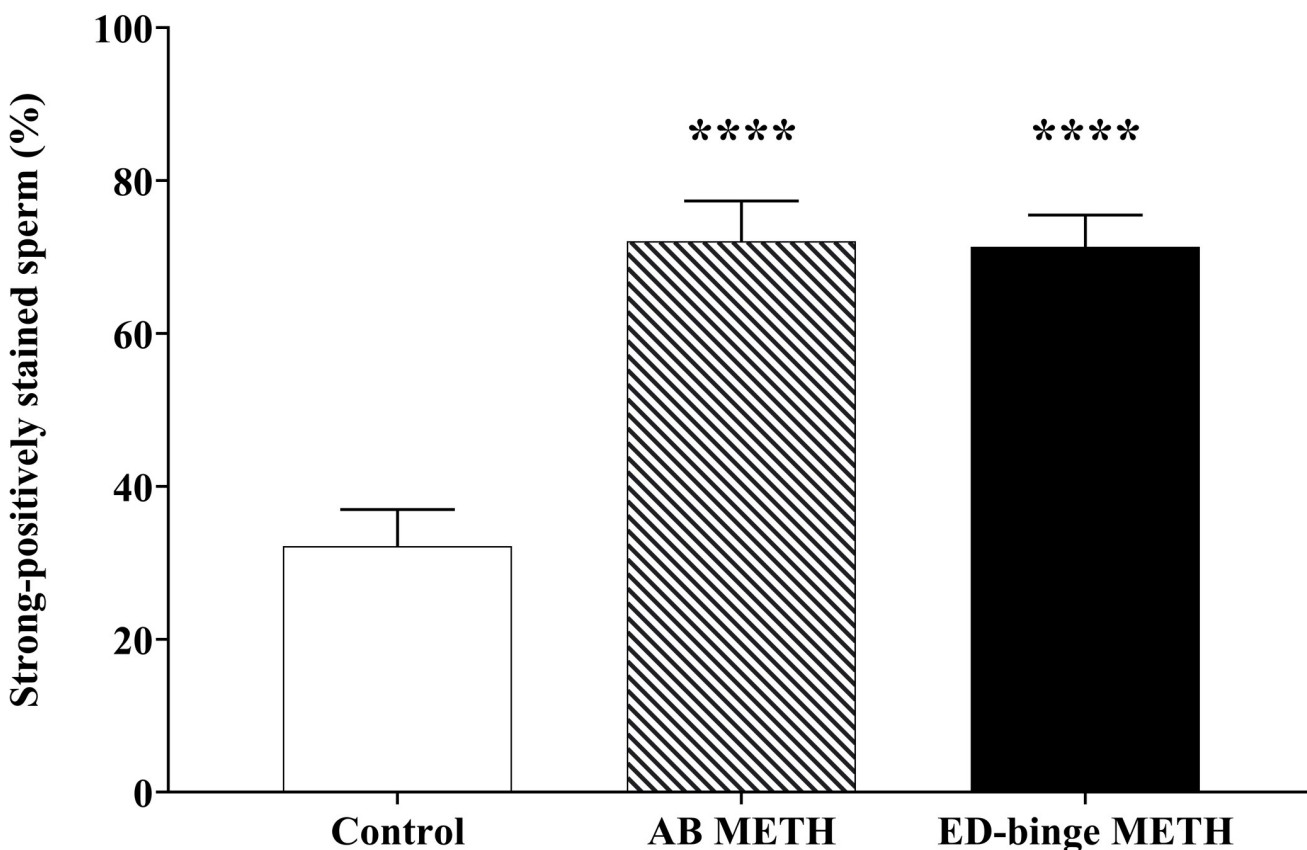

**Fig 5. Percentage of strong-positively stained sperm for GABA A-α1 receptor immunostaining (METH vs. control).** Values are shown as mean ± SEM, n = 8–9 per group (****P < 0.0001; Dunnett's post hoc test).

after METH exposure might be a compensatory effect to maintain testicular function. GABA A receptors, which act as chlorine ion selective ligand-gated ion channels, can be inhibited by L-type VDCC blockers [39]. Both GABA A receptors and T-type VDCCs are necessary for the secondary calcium oscillations that induce sperm capacitation and acrosome reaction [40]. GABA A receptor expression has been reported both in rat epididymal sperm and ejaculated human sperm [41–43]. In rat epididymal spermatozoa, its localisation was shown in the equatorial segment of the head, a region involved in acrosome reaction. Our result strongly supports the finding in the previous study, which found the immunoreaction of the GABA A-α1 receptor in the same location as in our study [17].

Interestingly, an increase in GABA A-α1 receptor was found in the epididymal spermatozoa of rats in the ED-binge METH group, which is in parallel with the change of GABA A-α1 receptor expression in the testis of these rats [6]. These findings indicate that in the METH-administered rats, the levels of GABA A-α1 receptor expression remain increased in the epididymal sperm after spermiation. Moreover, it has been reported that the rats in the ED-binge METH group, which were administered METH by mimicking its use in humans, had more severe poor sperm quality [5]. As shown in the correlation analysis results, the high levels of GABA A-α1 receptor expression in sperm were significantly related to the low percentage of normal sperm motility and morphology. These findings represent the responsiveness of

## GABA A-α1 receptor

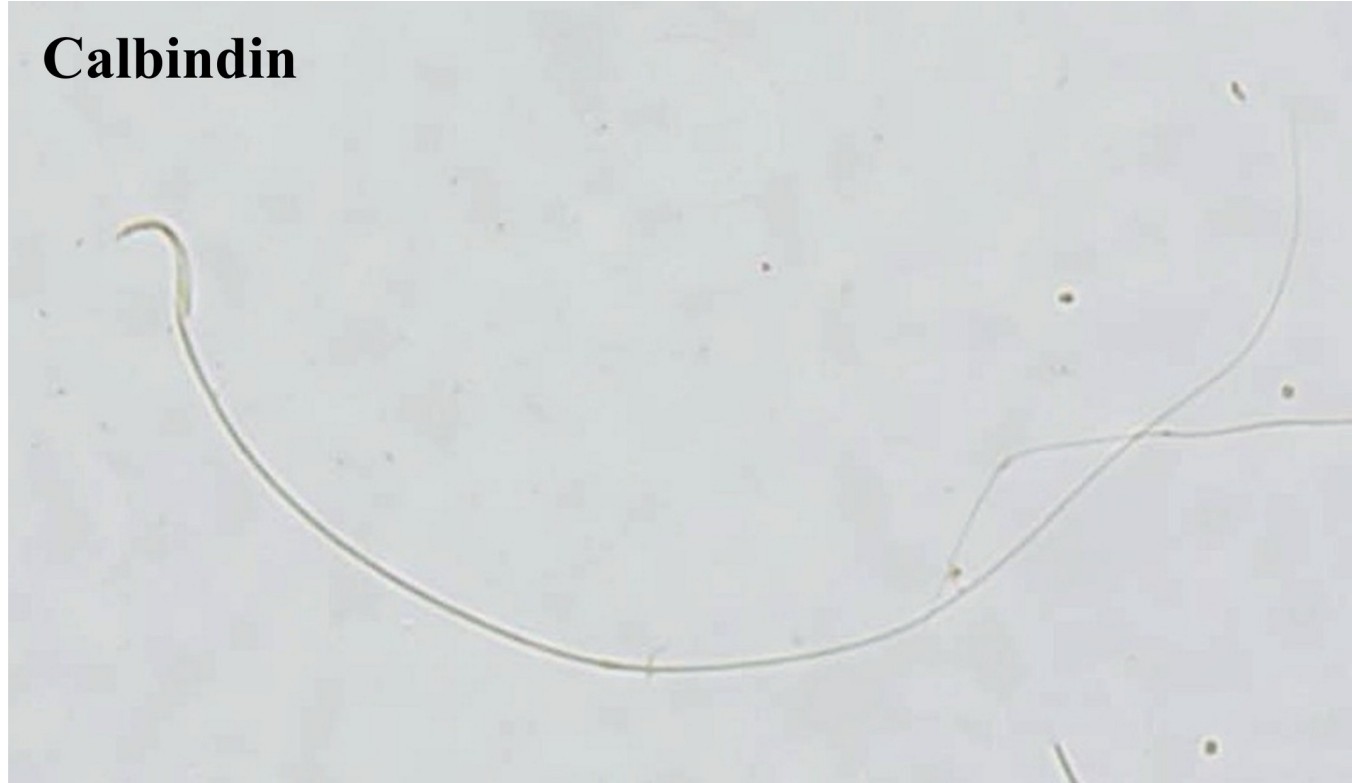

**Fig 6. Relative optical density of GABA A-α1 receptor expression levels in rat sperm (METH vs. control).** Values are shown as mean ± SEM, n = 8–9 per group (***$P < 0.001$; Dunnett's post hoc test).

**Fig 7. Positive immunostaining for calbindin in rat epididymal spermatozoa at 200× magnification.**

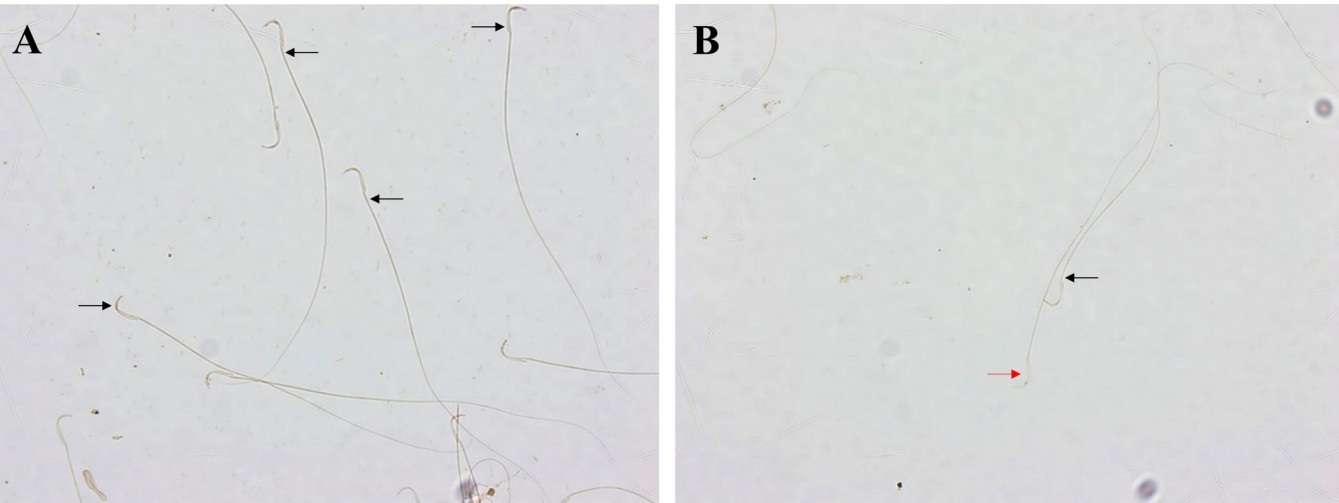

**Fig 8. Negative-to-strongly-positive immunostaining for calbindin in rat epididymal spermatozoa at 200× magnification.** (A, B) Black arrows indicate the strong-positively stained sperm. (B) A red arrow indicates a negatively stained spermatozoon.

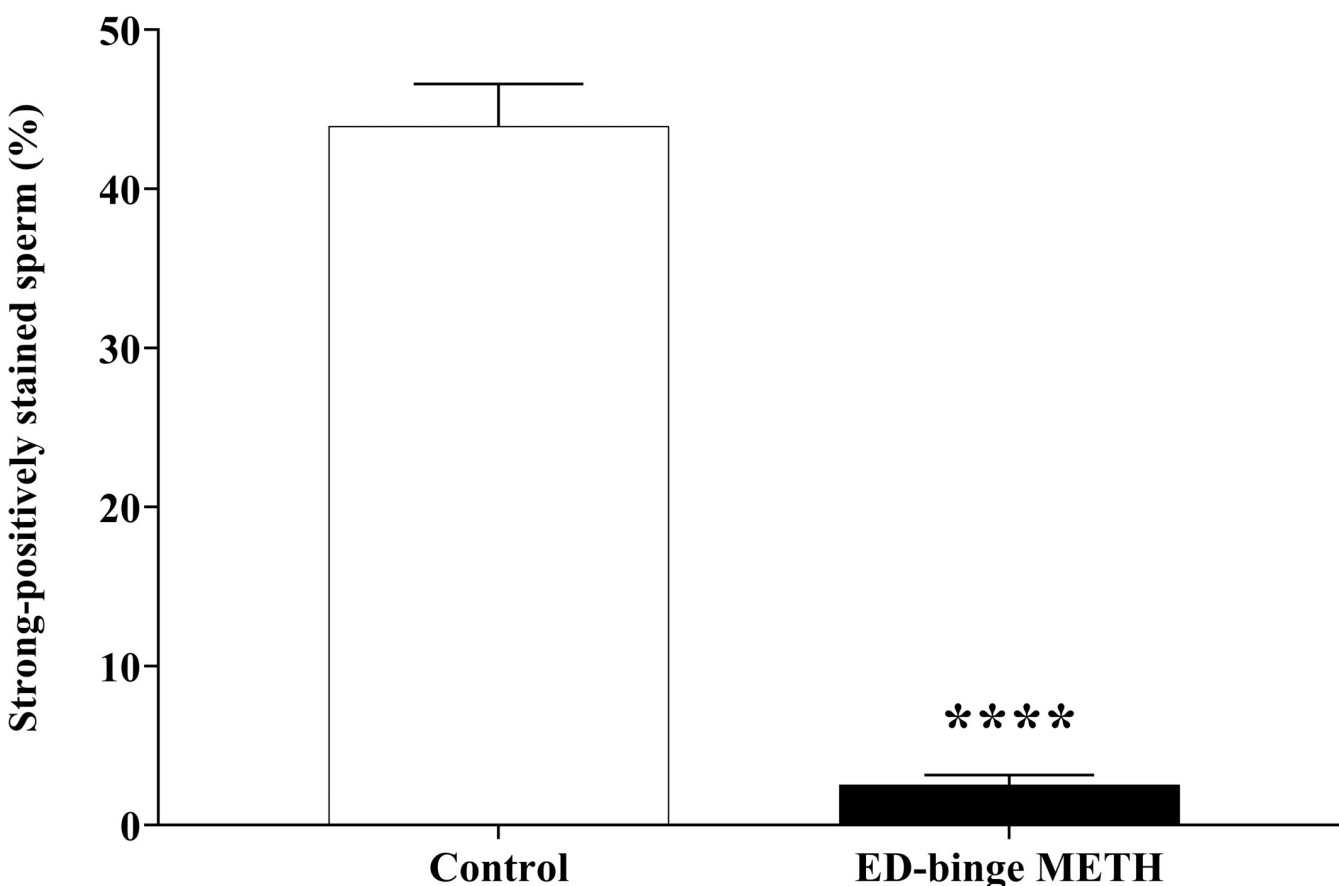

**Fig 9. Percentage of strong-positively stained sperm for calbindin immunostaining (METH vs. control).** Values are shown as mean ± SEM, n = 5 per group (****P < 0.0001; unpaired Student's t test).

# Calbindin

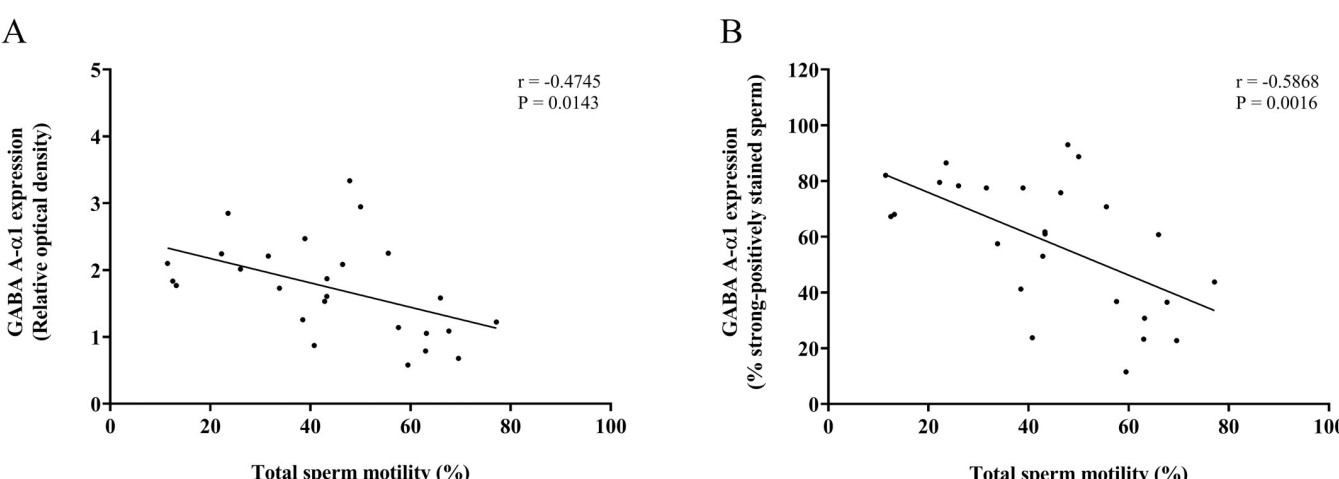

**Fig 10. Relative optical density of calbindin expression levels in rat sperm (METH vs. control).** Values are shown as mean ± SEM, n = 5 per group (****P < 0.0001; unpaired Student's t test).

**Fig 11. Correlations between the percentage of total sperm motility and GABA A-α1 receptor expression in sperm.** (A) Relative optical density. (B) The percentage of strong-positively stained sperm for GABA A-α1 receptor immunostaining. Linear regression line (black line) fitted to all data points.

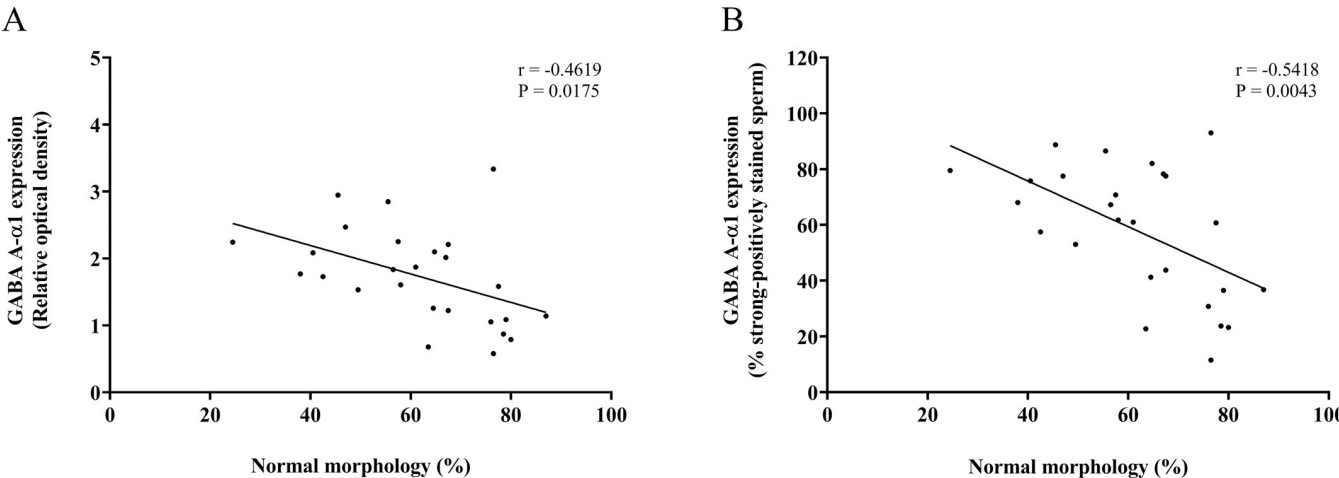

**Fig 12. Correlations between the percentage of normal sperm morphology and GABA A-α1 receptor expression in sperm.** (A) Relative optical density. (B) The percentage of strong-positively stained sperm for GABA A-α1 receptor immunostaining. Linear regression line (black line) fitted to all data points.

GABA relies on the severity of sperm quality impairment. The changes in VDCCs and GABA A-α1 receptor expression because of the adverse effects of METH also support their relationship in controlling the homeostatic regulation of calcium ion concentration in the testis and sperm. However, their molecular mechanisms in the control of testicular and sperm functions in fertilisation are not clear. Unfortunately, there are no results available in this study on the correlation of GABA A-α1 receptor expression in rat spermatozoa with other parameters of the fertilisation process such as capacitation or acrosome reaction. Moreover, we also found a decrease in Calb1 expression in the epididymal sperm of the METH-administered rat. Calb1 expression levels in the rat frontal cortex and hippocampus changed after METH exposure [44]. Calb1 belongs to the calcium-binding proteins (CaBPs) family, along with calmodulin, troponin C, and parvalbumin. CaBPs have been reported to localise in the sperm acrosome and principal piece of the sperm flagellum [45–47]. In the present study, Calb1 expression was also found in the acrosomal region and middle piece of the sperm. To act as a diffusible signal, calcium ions bind with CaBPs on the plasma membrane and in intracellular channels. The CaBPs such as calretinin and calmodulin regulate many biological functions required for sperm motility, capacitation, and the acrosome reaction [48–52]. They also regulate calcium influx signalling and VDCCs, which are implicated in human diseases [53, 54]. The alteration of CaBPs and VDCCs can result in the disruption of intracellular calcium homeostasis, leading to ROS production, cell cycle arrest, and apoptosis [55, 56]. In our result from proteomic analysis in the testis, the bioinformatics analysis of differentially expressed proteins of the calcium signalling pathway in METH-administered rats by the STITCH 5.0 database and the Search Tool for the Retrieval of Interacting Genes/Proteins (STRING) 11.0 database indicated that Cacna1d, Atp2b1, and Ntrk2 directly interact with Calb1. Therefore, abnormal calcium signalling in sperm promotes impaired sperm function and male infertility [57].

In conclusion, we have demonstrated the change of GABA A-α1 receptor expression in the sperm of METH-administered rats, which correlated with the impairment of sperm motility and morphology. Differential protein profiles in METH-administered rat testis were mostly classified in the calcium signalling pathway. These differentially expressed proteins might require for testicular and sperm functions and are associated with METH. Unfortunately, the change in the GABA A-α1 receptor protein expression is undetected by the proteomic analysis. It can be explained that the GABA A-α1 receptor protein expression in the testis is

specifically expressed in elongated sperm than other spermatogenic cells [58]. It might be the reason why the change in GABA A-α1 receptor protein expression cannot be detected in the testis from the proteomic study, but it can be mostly detected in epididymal sperm after spermiation. In this study, the validation of differentially expressed proteins in the calcium signalling pathway is limited to studies in the testis. However, validation of proteins in the calcium signalling pathway was done in epididymal sperm. A decrease in Calb1 expression was found in the sperm of METH-administered rats. We can suggest that the GABA A-α1 receptor in sperm might associate with proteins in the calcium signalling pathway to compensate for intracellular calcium homeostasis, maintaining testicular and sperm functions. The future study on the mechanism of the METH effect on testicular and sperm functions should focus primarily on proteins that are differentially expressed in the calcium signalling pathway associated with the GABA A-α1 receptor, especially Grm5, and VDCCs.

## Acknowledgments

We appreciate the facilities support from Naresuan University and the National Center for Genetic Engineering and Biotechnology, National Science and Technology Development Agency, Thailand.

## Author Contributions

**Conceptualization:** Sutisa Nudmamud-Thanoi, Samur Thanoi.

**Data curation:** Paweena Kaewman, Jitnapar Thongleart, Sawanya Charoenlappanit, Sittiruk Roytrakul.

**Formal analysis:** Paweena Kaewman, Sutisa Nudmamud-Thanoi, Jitnapar Thongleart, Sawanya Charoenlappanit, Sittiruk Roytrakul.

**Funding acquisition:** Sutisa Nudmamud-Thanoi, Samur Thanoi.

**Investigation:** Paweena Kaewman, Sutisa Nudmamud-Thanoi, Sittiruk Roytrakul, Samur Thanoi.

**Methodology:** Paweena Kaewman, Jitnapar Thongleart, Sawanya Charoenlappanit, Sittiruk Roytrakul.

**Project administration:** Samur Thanoi.

**Resources:** Sutisa Nudmamud-Thanoi, Sittiruk Roytrakul, Samur Thanoi.

**Supervision:** Sutisa Nudmamud-Thanoi, Samur Thanoi.

**Validation:** Paweena Kaewman.

**Writing – original draft:** Paweena Kaewman.

**Writing – review & editing:** Sutisa Nudmamud-Thanoi, Samur Thanoi.

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
