## [Decision Letter · Decision Letter 0]

2 Oct 2022

PONE-D-22-22951Differential protein expression of GABA A receptor alpha 1 subunit in rat spermatozoa associated with proteomic analysis in testis following methamphetamine administrationPLOS ONE

Dear Dr. Nudmamud-Thanoi,

Thank you for submitting your manuscript to PLOS ONE. After careful consideration, we feel that it has merit but does not fully meet PLOS ONE’s publication criteria as it currently stands. Therefore, we invite you to submit a revised version of the manuscript that addresses the points raised during the review process.

There are serious concerns on the validation of the observed results. You may note that one of the reviewers is of the view that this is a simple study wherein the effect of a chemical is being tested in the testicular tissue. The study lacks strong rationale. Further studies that provides strong evidence of the calcium signalling pathway are required. Further, the inclusion of positive control is a must. In light of the fact that the toxic effects of METH are well demonstrated in male reproduction, additional experiments are required so that this study contributes to the further understanding in this domain of research.

We look forward to receiving your revised manuscript.

Kind regards,

Suresh Yenugu

Academic Editor

PLOS ONE

Journal Requirements:

Reviewers' comments:

Reviewer's Responses to Questions

**Comments to the Author**

1. Is the manuscript technically sound, and do the data support the conclusions?

Reviewer #1: No

Reviewer #2: No

Reviewer #3: Yes

2. Has the statistical analysis been performed appropriately and rigorously? 

Reviewer #1: No

Reviewer #2: No

Reviewer #3: Yes

3. Have the authors made all data underlying the findings in their manuscript fully available?

Reviewer #1: Yes

Reviewer #2: No

Reviewer #3: Yes

4. Is the manuscript presented in an intelligible fashion and written in standard English?

Reviewer #1: No

Reviewer #2: Yes

Reviewer #3: Yes

5. Review Comments to the Author

Reviewer #1: I read the article with great interest. The study investigates the effect of Methamphetamine (METH) on spermatogenesis and work out the signalling involved in this alteration. I do not understand a clear hypothesis behind this study. Methamphetamine (METH) is one of the several chemicals that negatively affect spermatogenesis, a simple exposure followed by what molecular pathways get changed does not make the basis of an interesting study. It completely looks like the authors are studying the effect of a chemical compound on gene/protein expression in testis.

Reviewer #2: This study showed the differential expression of testicular proteins after methamphetamine exposure in rats. Methamphetamine (METH) is an addictive drug potentially affecting the male reproductive system. The study is important for discovering novel proteins, which might act as prognostic markers in infertile males exposed to METH. The study is important but not a novel study, as several groups have reported affect of METH in neuronal cells, brain, sertoli cells and rat testis. The METH disrupted testicular model is interesting but the phenotype of the testis with this dose is lacking in the manuscript. Some crucial points for this manuscript

1) Did the author check for other organs or primarily the brain or nervous system as a positive control for the METH induced testis model.

2)Validation of the mass spectrometry results for differentially expressed proteins need to be done by Western or IHC.

3) The authors highlight calcium signalling pathway as the topmost hit from KEGG analysis. It will be good to check the intracellular calcium levels of sperm to correlate the differential expression of proteins affecting Calcium pathway.

4) The authors also report increased staining in GABA-A receptor alpha 1 subunit for Meth treated sperm by Immunohistochemistry analysis.

LC MS data did not reveal any GABA receptor expression. How do the authors explain this

5) The work needs more refinement to understand how this dose of methamphetamine in testis is crossing the blood testis barrier and the regulatory proteins can be an important diagnostic markers in male infertility.

Reviewer #3: The present work entitled Differential protein expression of GABA A receptor alpha 1 subunit in rat spermatozoa

associated with proteomic analysis in testis following methamphetamine administration' describes that METH influences

intracellular calcium homeostasis by acting through the calcium signalling pathway associated proteins. Moreover, it might disrupt ion homeostasis in sperm through the GABA A-α1 receptor, triggering a change in intracellular calcium and chloride ions.

The present work though is interesting has some shortcomings which needs to be addressed.

1. Earlier reports have shown that METH decreases testicular functions, what is novel here?

2. GABA has been shown to be positively correlated with sperm functions, GABA facilitated the tyrosine phosphorylation of proteins in sperm, which is a marker of sperm capacitation. The present study shows an entire different angle. Justify.

3. The present study talks about calcium signaling but did not show any such validation data how METH experimentally affects the calcium signaling pathway?

4. The key pathways and markers pointed out from the proteomic study needs to be validated in testicular samples.

6. PLOS authors have the option to publish the peer review history of their article (what does this mean?). If published, this will include your full peer review and any attached files.

Reviewer #1: No

Reviewer #2: No

Reviewer #3: No

---

## [Author Response · Author response to Decision Letter 0]

23 Nov 2022

Reviewer #1: I read the article with great interest. The study investigates the effect of Methamphetamine (METH) on spermatogenesis and work out the signalling involved in this alteration. I do not understand a clear hypothesis behind this study. Methamphetamine (METH) is one of the several chemicals that negatively affect spermatogenesis, a simple exposure followed by what molecular pathways get changed does not make the basis of an interesting study. It completely looks like the authors are studying the effect of a chemical compound on gene/protein expression in testis.

Author response: 

• Many illicit drugs (e.g., methamphetamine, marijuana, cocaine, and anabolic-androgenic steroids (AAS)) have adverse effects on male infertility [1]. However, it is still unclear exactly how these illicit drugs work to impair testicular and sperm functions. Methamphetamine (METH) is an addictive drug whose adverse effects on the testis and sperm have been reported in several studies. Abuse of METH leads to neurotoxicity, addiction, and a change in the GABAergic system in the brain. The adverse effects of METH have been reported not only in the brain but also in the testicular and sperm functions. However, it is still unclear exactly how METH affects these functions. There is evidence that the GABAergic system has been found in the testis and sperm. The mechanisms and pathways of an addictive substance such as METH might involve the GABAergic system. In the previous study on the GABAergic system in the testis, we found an increase in GABA concentration and the mRNA expression of the GABA A-α1 receptor and GABA synthesising enzyme after METH exposure [2]. These results indicated that METH acts on the GABAergic system through the GABA A-α1 receptor. According to these findings, we believed that the GABA A-α1 receptor might be the key marker for defining testicular function. Moreover, it might interact with other pathways to control testicular function after exposure to METH. This has become one of the hypotheses of the present study. The proteomic analysis was done in the testis of METH-administered rats to verify this hypothesis. Another hypothesis is that the GABA A-α1 receptor in sperm might be important for the responsiveness of sperm function to METH effect as well. Therefore, the GABA A-α1 receptor protein expression in epididymal sperm was studied. The protein expression in the calcium signalling pathway, calbindin (Calb1), a member of the calcium-binding protein family, was studied in sperm after METH exposure. This result demonstrates the validation of the relationship between calcium signalling proteins in sperm and METH exposure. We expected that our results could support a role for the GABA A-α1 receptor in the mechanistic pathway of the METH effect. Medicines that specifically target GABAergic system components may be an alternative treatment for male infertility in drug users.

References

1. Fronczak CM, Kim ED, Barqawi AB. The insults of illicit drug use on male fertility. J Androl. 2012;33(4):515-28. doi:10.2164/jandrol.110.011874.1. 

2. Kaewman P, Nudmamud-Thanoi S, Thanoi S. GABAergic Alterations in the Rat Testis after Methamphetamine Exposure. Int J Med Sci. 2018;15(12):1349-54. doi: 10.7150/ijms.27609.

Reviewer #2: This study showed the differential expression of testicular proteins after methamphetamine exposure in rats. Methamphetamine (METH) is an addictive drug potentially affecting the male reproductive system. The study is important for discovering novel proteins, which might act as prognostic markers in infertile males exposed to METH. The study is important but not a novel study, as several groups have reported affect of METH in neuronal cells, brain, sertoli cells and rat testis. The METH disrupted testicular model is interesting but the phenotype of the testis with this dose is lacking in the manuscript. Some crucial points for this manuscript

1) Did the author check for other organs or primarily the brain or nervous system as a positive control for the METH induced testis model.

Author response:

• Yes, we studied several organs, especially the brain. The model of METH exposure that we used in this study was previously reported in the brain and addictive behavioural changes were also studied and confirmed. METH-administered rats revealed changes in components of the GABAergic and glutamatergic systems that were associated with neurotoxic effects [1-3]. Moreover, the protocol and antibody used in the immunocytochemistry analysis for detecting the expression of GABA A-α1 receptor and calbindin in sperm in this study were previously studied in the rat brain.

2) Validation of the mass spectrometry results for differentially expressed proteins need to be done by Western or IHC.

Author response:

• The change in protein expression of GABA A-α1 receptor in the epididymal sperm after methamphetamine (METH) exposure is the highlight of this study. The proteomic analysis in the testis was studied to support the hypothesis that, after METH exposure, the GABA A-α1 receptor might work together with other pathways to influence the function of the testis and the sperm. As we found, the calcium signalling pathway in the testis mostly responded to METH exposure. We absolutely agree that differentially expressed proteins in the calcium signalling pathway related to the GABA A-α1 receptor, especially metabotropic glutamate receptor 5 (Grm5) and voltage-dependent calcium ion channels (VDCCs), will be the most interesting to study in the testis and sperm of METH-administered rats in the future. However, this study may have limitations due to the lack of validation of these differentially expressed proteins in the testis. Lines 347-354 and lines 357-359 have this limitation and suggestion inserted, respectively.

3) The authors highlight calcium signalling pathway as the topmost hit from KEGG analysis. It will be good to check the intracellular calcium levels of sperm to correlate the differential expression of proteins affecting Calcium pathway.

Author response:

• As we mentioned earlier, we focused on the change in protein expression of the GABA A-α1 receptor in the mature sperm after METH exposure. The proteomic analysis was done in the testis of METH-administered rats to verify our hypothesis that the GABA A-α1 receptor might play a role with other pathways to regulate testicular and sperm function after METH exposure. Interestingly, there is evidence that METH causes changes in the mRNA expression of voltage-dependent calcium channels (e.g., sperm-specific cation channels) in the testis, which can affect the homeostasis of intracellular calcium concentration in the testis [4]. It can support our hypothesis that METH acts on both the GABAergic system through the GABA A-α1 receptor and the calcium signalling pathway in the testis. This study also included the expression of Calb1, a member of the calcium-binding proteins family, in sperm after METH exposure. Results are shown in lines 209-215 and Fig. 7-10. The discussion is in lines 327-342. A decrease in Calb1 expression in METH-administered rats can indicate the disruption of intracellular calcium homeostasis in sperm caused by METH exposure. The bioinformatics analysis also supports the relationship between the METH effect and the differentially expressed proteins of the calcium signalling pathway in METH-administered rats, including Calb1.

4) The authors also report increased staining in GABA-A receptor alpha 1 subunit for Meth treated sperm by Immunohistochemistry analysis. LC MS data did not reveal any GABA receptor expression. How do the authors explain this

Author response:

• According to our findings in the previous study, the mRNA expression of GABA A-α1 receptor in the testis was increased in METH-administered rats [5]. Because of this evidence, we believed that the GABA A-α1 receptor plays a role in the response to the METH effect in the testis. Unfortunately, the change in the GABA A-α1 receptor protein expression cannot be identified by the proteomic analysis as we had expected. It can be explained that the GABA A-α1 receptor protein expression in the testis is specifically expressed in elongated sperm (defining them as mature sperm) than other spermatogenic cells [6]. It might be the reason why the change in GABA A-α1 receptor protein expression cannot be detected in the testis from the proteomic analysis, but it can be mostly detected in epididymal sperm after spermiation. This explanation has been added to lines 347-351.

5) The work needs more refinement to understand how this dose of methamphetamine in testis is crossing the blood testis barrier and the regulatory proteins can be an important diagnostic markers in male infertility.

Author response:

• A review of previous research suggests that the pharmacology and toxicology of METH cause abuse potential and neurological effects. Several studies have also reported its effects on testicular and sperm functions [7]. The most popular pattern for METH abuse is the "binge" pattern: continuous use of multiple high doses to maintain the high METH concentration in the blood [8]. Interestingly, METH-administrated rats in an escalating dose‒binge pattern, which mimicked METH use in humans, had more severe poor sperm quality [9]. Absorption and distribution properties of METH administration can cause the accumulation of METH in target organs such as the brain and testis. METH can rapidly cross the blood-brain barrier by binding with transporters located on neuronal membranes such as dopamine, norepinephrine, and serotonin transporters [10]. Its lipophilic nature allows it to enter the cell more easily than other illicit drugs. Interestingly, those components of autonomic innervation were also found in the testis [11-12]. Furthermore, catecholamines, which include dopamine, epinephrine, and norepinephrine, play functional roles in testicular maturation and development. They interact with their receptors in the testis to stimulate testosterone production [13]. The catecholamine-synthesizing enzymes were also found in the testis [14]. Interestingly, our previous study indicated changes in catecholamine concentration in the testis of METH-administrated rats [15]. These rats also revealed the alteration of catecholamine receptors in the testis [16]. These findings indicate that METH might enter and have an impact on the testes via the catecholamine autonomic pathway. Although the mechanism for METH transport in the testis is not yet clarified, it is possible that METH can cross the blood-testis barrier in the same pattern as it crosses the blood-brain barrier in the brain. METH might bind to the transporters for entering the testicular membrane. Previous findings that METH might affect testicular function through neurotransmitter secretion also supported this hypothesis [17]. METH can have a strong impact on the activities of testicular cells and mature sperm through the responsiveness of the neurological pathway, especially the GABAergic system. In the present study, METH-administered rats revealed a change in the GABA A-α1 receptor protein expression in epididymal sperm that negatively correlated with sperm quality and a change in the proteomic profiles in the testis. The mRNA expression of GABA A-α1 receptor in the testis was altered after METH exposure [5]. For this reason, the GABA A-α1 receptor can be an important marker for the diagnosis of spermatogenesis and sperm quality.

References

1. Kerdsan W, Thanoi S, Nudmamud-Thanoi S. Changes in glutamate/NMDA receptor subunit 1 expression in rat brain after acute and subacute exposure to methamphetamine. J Biomed Biotechnol. 2009;2009:329631. doi: 10.1155/2009/329631.

2. Kerdsan W, Thanoi S, Nudmamud-Thanoi S. Changes in the neuronal glutamate transporter EAAT3 in rat brain after exposure to methamphetamine. Basic Clin Pharmacol Toxicol. 2012;111(4):275-8. doi:10.1111/j.1742-7843.2012.00899.x

3. Veerasakul S, Thanoi S, Reynolds GP, Nudmamud-Thanoi S. Effect of Methamphetamine Exposure on Expression of Calcium Binding Proteins in Rat Frontal Cortex and Hippocampus. Neurotox Res. 2016;30(3):427-33. doi:10.1007/s12640-016-9628-2.

4. Allaeian Jahromi Z, Meshkibaf MH, Naghdi M, Vahdati A, Makoolati Z. Methamphetamine Downregulates the Sperm-Specific Calcium Channels Involved in Sperm Motility in Rats. ACS Omega. 2022;7(6):5190-96. doi: 10.1021/acsomega.1c06242.

5. Kaewman P, Nudmamud-Thanoi S, Thanoi S. GABAergic Alterations in the Rat Testis after Methamphetamine Exposure. Int J Med Sci. 2018;15(12):1349-54. doi: 10.7150/ijms.27609.

6. Geigerseder C, Doepner R, Thalhammer A, Frungieri MB, Gamel-Didelon K, Calandra RS, et al. Evidence for a GABAergic system in rodent and human testis: local GABA production and GABA receptors. Neuroendocrinology. 2003;77(5):314-23. doi: 10.1159/000070897.

7. Fronczak CM, Kim ED, Barqawi AB. The insults of illicit drug use on male fertility. J Androl. 2012;33(4):515-28. doi:10.2164/jandrol.110.011874. 

8. Cho AK, Melega WP. Patterns of methamphetamine abuse and their consequences. J Addict Dis. 2002;21(1):21-34. doi: 10.1300/j069v21n01_03.

9. Nudmamud-Thanoi S, Sueudom W, Tangsrisakda N, Thanoi S. Changes of sperm quality and hormone receptors in the rat testis after exposure to methamphetamine. Drug Chem Toxicol. 2016;39(4):432-8. doi: 10.3109/01480545.2016.1141421.

10. Rusyniak DE. Neurologic manifestations of chronic methamphetamine abuse. Neurol Clin. 2011;29(3):641-55. doi:10.1016/j.ncl.2011.05.004. 

11. Jiménez-Trejo F, Coronado-Mares I, Arriaga-Canon C, et al. Indolaminergic System in Adult Rat Testes: Evidence for a Local Serotonin System. Front Neuroanat. 2021;14:570058. doi:10.3389/fnana.2020.570058.

12. González CR, González B, Matzkin ME, Muñiz JA, Cadet JL, Garcia-Rill E, Urbano FJ, Vitullo AD, Bisagno V. Psychostimulant-Induced Testicular Toxicity in Mice: Evidence of Cocaine and Caffeine Effects on the Local Dopaminergic System. PLoS One. 2015;10(11):e0142713. doi: 10.1371/journal.pone.0142713.

13. Mayerhofer A, Steger RW, Gow G, Bartke A. Catecholamines stimulate testicular testosterone release of the immature golden hamster via interaction with alpha- and beta-adrenergic receptors. Acta Endocrinol (Copenh). 1992;127(6):526-30. doi: 10.1530/acta.0.1270526.

14. Davidoff MS, Ungefroren H, Middendorff R, Koeva Y, Bakalska M, Atanassova N, et al. Catecholamine-synthesizing enzymes in the adult and prenatal human testis. Histochem Cell Biol. 2005 Sep;124(3-4):313-23. doi: 10.1007/s00418-005-0024-x.

15. Janphet S, Nudmamud-Thanoi S, Thanoi S. Alteration of catecholamine concentrations in rat testis after methamphetamine exposure. Andrologia. 2017 Mar;49(2). doi: 10.1111/and.12616. Epub 2016 May 11. PMID: 27167778.

16. Thanoi S, Janphet S, Nudmamud-Thanoi S. Changes of dopamine D2, alpha1 adrenergic receptor expressions and developmental stages of seminiferous tubule in rat testis after methamphetamine administration: A preliminary study. Songklanakarin Journal of Science & Technology. 2020;42(4).

17. Alavi SH, Taghavi MM, Moallem SA. Evaluation of effects of methamphetamine repeated dosing on proliferation and apoptosis of rat germ cells. Syst Biol Reprod Med. 2008;54(2):85-91. doi:10.1080/19396360801952078. 

Reviewer #3: The present work entitled Differential protein expression of GABA A receptor alpha 1 subunit in rat spermatozoa

associated with proteomic analysis in testis following methamphetamine administration' describes that METH influences

intracellular calcium homeostasis by acting through the calcium signalling pathway associated proteins. Moreover, it might disrupt ion homeostasis in sperm through the GABA A-α1 receptor, triggering a change in intracellular calcium and chloride ions.

The present work though is interesting has some shortcomings which needs to be addressed.

1. Earlier reports have shown that METH decreases testicular functions, what is novel here?

Author response:

• Several studies have reported the adverse effects of methamphetamine (METH) on sperm quality and testicular function. It is still unclear exactly how METH decreases testicular function. In our previous study on the GABAergic system in the testis, we found an increase in GABA concentration and the mRNA expression of the GABA A-α1 receptor and GABA synthesising enzyme. According to this conclusion, we thought that the GABA A-α1 receptor might interact with other pathways to control testicular function after exposure to METH. The present study reported novel evidence about the responsiveness of proteins in the calcium signalling pathway to METH exposure. Moreover, the highlight of this study is the change in GABA A-α1 receptor protein expression in epididymal sperm after METH exposure, which was reported for the first time. The negative correlation between GABA A-α1 receptor protein expression in sperm and sperm quality, including sperm motility and morphology, was also reported.

2. GABA has been shown to be positively correlated with sperm functions, GABA facilitated the tyrosine phosphorylation of proteins in sperm, which is a marker of sperm capacitation. The present study shows an entire different angle. Justify.

Author response:

• A previous study has revealed that GABA can induce sperm capacitation through a specific GABA A receptor by an intracellular mechanism depending on calcium influx and cAMP accumulation [1]. This mechanism is controlled by the tyrosine phosphorylation of sperm proteins [2]. Supporting evidence for the mechanism of GABA on sperm capacitation and hyperactivation indicates that GABA plays a role in the changes in calcium, chloride, and bicarbonate ions via GABA A receptor [3]. GABA A receptor also controls calcium signalling in the modulation of sperm kinetic parameters (including sperm motility) [4]. The role of GABA on sperm functions (e.g., capacitation, acrosome reaction, hyperactivation) has been reported, not only as an excitatory effect but also as an inhibitory effect, by acting mostly through the GABA A receptors. Several factors, such as progesterone, serotonin, and GABA, can act as inducers of those sperm functions. Interestingly, both GABA and progesterone can act through the GABA A receptor in the sperm to induce sperm functions [5-6]. Competitive binding of GABA to GABA A receptor can inhibit progesterone-enhanced hyperactivation [5]. This finding indicates an inhibitory effect of GABA. All of these findings indicate that it is still unclear exactly how GABA works to control these sperm functions. In the present study, high levels of GABA A‒α1 receptor protein expression in epididymal sperm were found in METH-administered rats. As shown in the correlation analysis results, it was significantly related to the low normal sperm motility and morphology. It is possible that after METH exposure, the increase in GABA A‒α1 receptor occurs in the sperm to compensate for the sperm impairment and maintain the homeostasis of sperm function.

3. The present study talks about calcium signaling but did not show any such validation data how METH experimentally affects the calcium signaling pathway?

Author response:

• The key finding of this study is the change in GABA A-α1 receptor protein expression in epididymal sperm resulting from METH exposure. We also reported a negative correlation between GABA A-α1 receptor protein expression and sperm quality (including motility and morphology), as shown in our results. Interestingly, these results related to the change of the GABA A-α1 receptor mRNA expression in the testis of METH-administered rats, as previously reported [7]. Moreover, we also believed that the GABA A-α1 receptor might play a role with other pathways to regulate testicular and sperm function resulting from exposure to METH. In order to verify this hypothesis, the proteomic analysis of the testis of METH-administered rats was investigated. The results from the proteomic analysis indicated that proteins in the KEGG pathway belonging to the calcium signalling pathway are the most interesting to be pointed out. Although we do not have the validation result of proteins in the calcium signalling pathway in the testis, we provide information in the discussion to indicate the association of these proteins with the change in GABA A-α1 receptor of METH-administered rats. It can be suggested that future studies on the testis and sperm of METH-administered rats should focus primarily on proteins that are differentially expressed in the calcium signalling pathway associated with the GABA A-α1 receptor, especially metabotropic glutamate receptor 5 (Grm5) and voltage-dependent calcium ion channels (VDCCs). This suggestion has been added to lines 357-359. This study also included the validation of a protein in the calcium signalling pathway, calbindin (Calb1), a member of the calcium-binding protein (CaBP) family, in sperm after METH exposure. Results are shown in lines 209-215 and Fig. 7-10. The discussion is in lines 327-342. The alteration of CaBPs and VDCCs can result in disruption of intracellular calcium homeostasis, leading to ROS production, cell cycle arrest, and apoptosis [8-9]. For this reason, abnormal calcium signalling in sperm promotes impaired sperm function and male infertility [10]. A decrease in Calb1 expression in METH-administered rats can indicate the disruption of intracellular calcium homeostasis in sperm caused by METH exposure. The bioinformatics analysis also supports the relationship between the METH effect and the differentially expressed proteins of the calcium signalling pathway in METH-administered rats, including Calb1.

4. The key pathways and markers pointed out from the proteomic study needs to be validated in testicular samples.

Author response:

• Because of the finding of GABAergic system change in the testis of METH-administered rats [7], it was predicted that the key marker for defining testicular and sperm functions might be the GABA A-α1 receptor. Unfortunately, the results from the proteomic analysis in the testis cannot support the previous finding about the change of the GABA A-α1 receptor mRNA expression as we had expected. There is evidence that the GABA A-α1 receptor protein expression in the testis is specifically expressed in elongated sperm than other spermatogenic cells [11]. It might be the reason why the change in GABA A-α1 receptor protein expression cannot be found in the testis from the proteomic study, but it can be detected in mature sperm after spermiation. Therefore, we did not validate the protein expression of the GABA A-α1 receptor in testis but focused on its expression in epididymal sperm. This is perhaps the limitation of this study. This limitation has been added to lines 313-319.

References

1. Ritta MN, Bas DE, Tartaglione CM. In vitro effect of gamma-aminobutyric acid on bovine spermatozoa capacitation. Mol Reprod Dev. 2004;67(4):478-86. doi: 10.1002/mrd.20038.

2. Kurata S, Hiradate Y, Umezu K, Hara K, Tanemura K. Capacitation of mouse sperm is modulated by gamma-aminobutyric acid (GABA) concentration. J Reprod Dev. 2019;65(4):327-34. doi: 10.1262/jrd.2019-008.

3. Jin JY, Chen WY, Zhou CX, Chen ZH, Yu-Ying Y, Ni Y, et al. Activation of GABAA receptor/Cl- channel and capacitation in rat spermatozoa: HCO3- and Cl- are essential. Syst Biol Reprod Med. 2009;55(2):97-108. doi: 10.1080/19396360802626648.

4. Calogero AE, Hall J, Fishel S, Green S, Hunter A, D'Agata R. Effects of gamma-aminobutyric acid on human sperm motility and hyperactivation. Mol Hum Reprod. 1996;2(10):733-8. doi: 10.1093/molehr/2.10.733.

5. Kon H, Takei GL, Fujinoki M, Shinoda M. Suppression of progesterone-enhanced hyperactivation in hamster spermatozoa by γ-aminobutyric acid. J Reprod Dev. 2014;60(3):202-9. doi: 10.1262/jrd.2013-076.

6. Calogero AE, Burrello N, Ferrara E, Hall J, Fishel S, D'Agata R. Gamma-aminobutyric acid (GABA) A and B receptors mediate the stimulatory effects of GABA on the human sperm acrosome reaction: interaction with progesterone. Fertil Steril. 1999;71(5):930-6. doi: 10.1016/s0015-0282(99)00063-1.

7. Kaewman P, Nudmamud-Thanoi S, Thanoi S. GABAergic Alterations in the Rat Testis after Methamphetamine Exposure. Int J Med Sci. 2018;15(12):1349-54. doi: 10.7150/ijms.27609.

8. Schwab BL, Guerini D, Didszun C, Bano D, Ferrando-May E, Fava E, et al. Cleavage of plasma membrane calcium pumps by caspases: a link between apoptosis and necrosis. Cell Death Differ. 2002;9(8):818-31. doi: 10.1038/sj.cdd.4401042.

9. Cross JL, Meloni BP, Bakker AJ, Lee S, Knuckey NW. Modes of Neuronal Calcium Entry and Homeostasis following Cerebral Ischemia. Stroke Res Treat. 2010;2010:316862. doi: 10.4061/2010/316862.

10. Rahman MS, Kwon WS, Pang MG. Calcium influx and male fertility in the context of the sperm proteome: an update. Biomed Res Int. 2014;2014:841615. doi:10.1155/2014/841615.

11. Geigerseder C, Doepner R, Thalhammer A, Frungieri MB, Gamel-Didelon K, Calandra RS, et al. Evidence for a GABAergic system in rodent and human testis: local GABA production and GABA receptors. Neuroendocrinology. 2003;77(5):314-23. doi: 10.1159/000070897.

---

## [Decision Letter · Decision Letter 1]

19 Dec 2022

Differential protein expression of GABA A receptor alpha 1 subunit and calbindin in rat spermatozoa associated with proteomic analysis in testis following methamphetamine administration

PONE-D-22-22951R1

Dear Dr. Nudmamud-Thanoi,

We’re pleased to inform you that your manuscript has been judged scientifically suitable for publication and will be formally accepted for publication once it meets all outstanding technical requirements.

Kind regards,

Suresh Yenugu

Academic Editor

PLOS ONE

Additional Editor Comments (optional):

Reviewers' comments:

Reviewer's Responses to Questions

**Comments to the Author**

1. If the authors have adequately addressed your comments raised in a previous round of review and you feel that this manuscript is now acceptable for publication, you may indicate that here to bypass the “Comments to the Author” section, enter your conflict of interest statement in the “Confidential to Editor” section, and submit your "Accept" recommendation.

Reviewer #1: All comments have been addressed

Reviewer #2: All comments have been addressed

Reviewer #3: All comments have been addressed

2. Is the manuscript technically sound, and do the data support the conclusions?

Reviewer #1: Yes

Reviewer #2: No

Reviewer #3: Yes

3. Has the statistical analysis been performed appropriately and rigorously? 

Reviewer #1: Yes

Reviewer #2: N/A

Reviewer #3: Yes

4. Have the authors made all data underlying the findings in their manuscript fully available?

Reviewer #1: Yes

Reviewer #2: Yes

Reviewer #3: Yes

5. Is the manuscript presented in an intelligible fashion and written in standard English?

Reviewer #1: Yes

Reviewer #2: Yes

Reviewer #3: Yes

6. Review Comments to the Author

Reviewer #1: All my comments have been addressed. The manuscript looks good now and may now be accepted. I have no further comments.

Reviewer #2: The authors have clearly answered to all the comments. However, the study has limitations due to the lack of validation of these differentially expressed proteins in the testis as mentioned by the authors. This study can be a preliminary understanding and basis for GABA A-α1 receptor in the pathway of the METH effect in testis.

Reviewer #3: Authors have satisfactorily taken care of all comments raised. The manuscript entitled 'Differential protein expression of GABA A receptor alpha 1 subunit and calbindin in rat spermatozoa associated with proteomic analysis in testis following methamphetamine administration' in the present form can be accepted.

7. PLOS authors have the option to publish the peer review history of their article (what does this mean?). If published, this will include your full peer review and any attached files.

Reviewer #1: **Yes: **Rajender Singh

Reviewer #2: **Yes: **Barnali Biswas

Reviewer #3: No

---

## [Editor Report · Acceptance letter]

26 Dec 2022

PONE-D-22-22951R1 

Differential protein expression of GABA A receptor alpha 1 subunit and calbindin in rat spermatozoa associated with proteomic analysis in testis following methamphetamine administration 

Dear Dr. Nudmamud-Thanoi:

I'm pleased to inform you that your manuscript has been deemed suitable for publication in PLOS ONE. Congratulations! Your manuscript is now with our production department. 

Kind regards, 

on behalf of

Dr. Suresh Yenugu 

Academic Editor

PLOS ONE